# Effects of Aquatic Exercises for Women with Rheumatoid Arthritis: A 12-Week Intervention in a Quasi-Experimental Study with Pain as a Mediator of Depression

**DOI:** 10.3390/ijerph20105872

**Published:** 2023-05-19

**Authors:** Miguel A. Perez-Sousa, Jéssica Pedro, Rocio Carrasco-Zahinos, Armando Raimundo, Jose A. Parraca, Pablo Tomas-Carus

**Affiliations:** 1Department of Physical Education, Faculty of Education, University of Córdoba, 14071 Cordoba, Spain; 2Epidemiology of Physical Activity and Fitness across Lifespan Research Group, University of Seville, 41004 Seville, Spain; 3Departamento de Desporto e Saúde, Escola de Saúde e Desenvolvimento Humano, Universidade de Évora, 7004-516 Evora, Portugal; 4Faculty of Education, Psychology and Sport Sciences, University of Huelva, Avenida de las Fuerzas Armadas s/n, 21007 Huelva, Spain; 5Comprehensive Health Research Centre (CHRC), University of Évora, 7004-516 Evora, Portugal

**Keywords:** aquatic exercise, rheumatoid arthritis, physical fitness, depression, pain

## Abstract

Background: Rheumatoid arthritis (RA) is characterized by low physical fitness, pain, and depression. The present study aimed to examine the effects of a supervised aquatic exercise program on physical fitness, depression, and pain in women with RA and determine whether decreases in pain mediate depression. Methods: Forty-three women with RA, divided into an experimental group (EG; n = 21) and a control group (CG; n = 23), participated in a 12-week exercise program. Treatment effects were calculated via standardized difference or effect size (ES) using ANCOVA adjusted for baseline values (ES, 95% confidence interval (CI)). A simple panel of mediation was executed to determine whether changes in pain mediated improvements in depression after controlling for confounding variables, such as age, physical activity, and body mass index (BMI). Results: The aquatic exercise program had trivial and small effects on physical fitness, large effects on pain, and moderate effects on depression. The mediation model confirmed the indirect effect of pain on the decrease of depression in the participants of the aquatic exercise program. Conclusions: Participants with RA in the aquatic exercise program experienced improvements in physical fitness, depression, and joint pain. Moreover, the improvements in joint pain mediated improvements in depression.

## 1. Introduction

Rheumatoid arthritis (RA) is a clinical syndrome characterized by several inflammatory cascades and autoantibodies that lead to synovial inflammation related to damage in articular cartilage and the underlying bone [1]. It seems that 50% of the risk of developing RA is attributable to genetic factors [2]. Other associated risk factors are related to lifestyle, such as smoking and alcohol intake, oral contraceptive consumption, and low socioeconomic status [3,4]. The prevalence of RA is 0.5–1% of adults in developed countries, where three out of every four are women [5]. Pain is the most common symptom related to RA [6], which is usually accompanied by fatigue [7], loss of functional capacity, and a decline in quality of life [6].

As an alternative to pharmacological treatments for RA patients, physical exercise is considered an effective therapy to manage or improve RA symptoms. In this sense, profuse physical exercise programs have been implemented among RA patients, obtaining beneficial effects for functional capacity, perceived pain, and quality of life [8]. Within the broad typology of physical exercise, strengthening and aerobic exercises on land are the most common and are well-documented for RA [8]. However, aquatic exercises are also used as therapy in the treatment of inflammatory diseases [9,10]. Water brings physical properties, such as buoyancy, resistance and hydrostatic pressure, the promotion of muscle relaxation, and the unloading of joints. These characteristics encourage exercise practice, particularly in patients with rheumatic diseases.

Previous aquatic exercise programs in RA patients have shown improvements in fitness, pain, and depression [8,9,11]. From our point of view, the relationship between physical exercise, pain, and depression should be studied deeply. In this sense, it is well documented that patients with RA have a two to three times higher risk rate for developing depression [12]. Conversely, physical exercise decreases perceptions of bodily pain [13,14] and depression levels [11]. Therefore, the relationships between these three variables and the underlying mechanisms are unclear. Some researchers have found a mediated mechanism of depression influencing by bodily pain [15,16]. However, recent studies have indicated that, behind depression, there are several underlying mechanisms influencing pain [17,18]. Therefore, it is necessary to explore the role of pain as a mediator between depression and RA. To answer this question, we hypothesized that the female RA patients participating in the aquatic exercise program would experience decreases in pain and depression and improve physical fitness. Secondly, we hypothesized that the improvements in depression associated with participation in the program would be caused by reduced perceptions of pain.

Consequently, the aim of this study was twofold: (i) to investigate the effects of a supervised aquatic exercise program on physical fitness, depression, and pain in women with RA and (ii) to determine whether the underlying mechanism between participating in the program and depression was the decline of pain.

## 2. Materials and Methods

### 2.1. Design and Participants

The current research was conducted via a quasi-experimental study. An aquatic exercise intervention was developed for women with RA to compare the effects of participating in an aquatic exercise program with a control group over 12 weeks. The EG consisted of referrals to an exercise program, detailed below, while the CG continued with their normal life after being evaluated at the same time as the EG.

The study included a convenience sample recruited from a local hospital in Portugal. To be included in the study, participants had to meet the following criteria: aged 18–80 years; no participation in a physical exercise program or no history of physical exercise in the last three months; no severe comorbidities or self-reported contraindications for physical activity; having been diagnosed with RA by the physician. 

Of the 73 women initially contacted, 29 were excluded due to not meeting the inclusion criteria, and five desisted from participating in the study. Finally, 43 were enrolled for participation in the exercise program. The sample was randomly divided, by convenience, into either the EG (n = 21) or the CG (n = 23).

### 2.2. Measurements

#### 2.2.1. Anthropometrics Measurement

Height and body weight were measured using a portable stadiometer (SECA 213, Hamburg, Germany) and an electronic scale (Kendall graduated platform scale). BMI was calculated as weight in kilograms divided by the square of height in meters.

#### 2.2.2. Physical Fitness

It was administered as a standardized physical fitness battery for adults, whose activity duration was approximately 30 min. A warm-up exercise preceded the test by 5 min, including walking and mobility. The following fitness outcomes were then measured:

*Upper body strength*: Handgrip strength was measured using a dynamometer (TKK 5401; Takei Scientific Instruments Co., Ltd., Tokyo, Japan). Grip strength was measured in each hand with a 1 min rest interval, and the highest score of the six measurements was recorded.

*Lower body strength*: The 30 s chair stand test was used to assess lower body strength, following the standardized protocol [19].

*Upper and lower body flexibility*: Lower body flexibility was measured using the seated sit and reach test developed by *Jones* et al. [19]. The score was the number of centimeters short of reaching the toes (minus score) or reaching beyond the toes (plus score). The best score of two test trials was used to evaluate performance.

*Balance and agility*: Dynamic balance and agility were measured using the Timed Up and Go Test (TUG) along a 2.44 m corridor [20], recording the best score of two trials.

#### 2.2.3. Pain

We determined pain through a visual analog scale (VAS), which consisted of a 10 cm line, with two endpoints representing 0 (‘no pain’) and 10 (‘pain as bad as it could be’) [21].

#### 2.2.4. Depression

The Portuguese version of the Beck Depression Inventory-II (BDI-II) [22] was used to assess the severity of depressive symptomatology. The questionnaire includes questions about body image, hypochondriasis, difficulty working, sleep, and appetite loss.

#### 2.2.5. Physical Activity

Self-reported physical activity was ascertained using the Portuguese short version of the IPAQ [23], which assess physical activity performed in four domains: work, leisure, homework, and going to the job.

#### 2.2.6. Intervention

Those women in the intervention group participated in 45′ of aquatic exercises two times a week for three months, totaling 24 training sessions. The session was completed in small groups of 10–11 subjects in a pool heated to 30–32° degrees with depths of 1.2–1.5 m. The sessions consisted of three parts: the warm-up (10 min), the main set (30 min), and the cool-down (5 min). A graduate, with experience in aquatic training and rehabilitation, carried out training in sports sciences.

#### 2.2.7. Data Analysis

The Kolmogorov–Smirnov test with the Lillifort correction was initially used to test the data’s normality. Between-group differences in participants’ baseline characteristics were tested using the One-way ANOVA test. The treatment effects were calculated via standardized differences or effect sizes (ESs) using ANCOVA adjusted for baseline characteristics (ES, 95% confidence interval (CI)). The treatment effects (and 95% CI) were shown as the percentage of change relative to the initial status of the EG minus the percentage of change relative to the initial status of the CG (ΔEG − ΔCG). The percentage of change relative to the initial status was calculated using the compute variable function in the statistical package SPSS: [(Variable_12-week_ − Variable_baseline_)/Variable_baseline_] × 100. Effect sizes were measured using eta squared (η^2^).

Additionally, for a better comparison with other studies, all effect sizes, η^2^, were transformed into Cohen’s *d* standardized effect size units using a computation program (www.psychometrica.de, accessed on 6 March 2023) [24]. Meaningful inferences about magnitudes were made [25]. Next, the PROCESS macro for SPSS version 4.1 [26] was used to determine the indirect effect of participation in the aquatic program on depression through improvements in pain perception using mediation analysis. The indirect effect of the aquatic program (mediated by changes in pain) on changes in depression was estimated, while accounting for the direct effects the aquatic program had on decreases in pain and depression, after controlling for confounding variables, such as age, physical activity, and body mass index. Mediation hypotheses were analyzed using the bias-corrected bootstrap method with 5000 samples to calculate confidence intervals (95%). An indirect effect was considered significant when the confidence interval did not include zero. Statistical analyses were performed using the SPSS v.25 statistical package (IBM, New York, NY, USA). For all tests, the significance level was set at *p* < 0.05.

## 3. Results

Characteristics of study participants are shown in Table 1. Statistically significant differences were not found for age (years) (EG: 56.7 ± 11.2; CG: 56.2 ± 10.8; *p* = 0.892), weight (kg) (EG: 69.1 ± 7.7; CG: 69.7 ± 9.5; *p* = 0.826), or BMI (kg/m^2^) (EG: 27.1 ± 3.2; CG: 27.9 ± 3.5; *p* = 0.399).

Table 2 shows the effects of participating in an aquatic exercise program for 12 weeks for RA patients. The exercise program had positive effects on physical fitness, especially on upper body strength, with a percentage increase of 8.7%. Further, Cohen’s *d* effect sizes (and 95% CI); the percentage of chance for harmful, trivial, and beneficial outcomes; the clinical magnitude-based inference; descriptive statistics at baseline; the percentage of relative change compared to initial status; and the *p*-values of physical fitness were as follows: upper body strength (F_(1.44)_ = 25.5; *p* < 0.01; η^2^ = 0.378), lower body strength (F_(1.44)_ = 10.4; *p* < 0.01; η^2^ = 0.198), upper body flexibility (F_(1.44)_ = 0.72; *p* > 0.05; η^2^ = 0.018), lower body flexibility (F_(1.44)_ = 0.41; *p* > 0.05; η^2^ = 0.012), agility/dynamic balance (F_(1.44)_ = 16.7; *p* < 0.01; η^2^ = 0.285); joint pain–VAS (F_(1.44)_ = 33.23; *p* < 0.01; η^2^ = 0.442); and Beck depression inventory II scores (F_(1.44)_ = 33.53; *p* < 0.01; η^2^ = 0.444).

Figure 1 shows the meditation model used to determine whether the improvement in joint pain could mediate the change in depression after participating in the aquatic exercise program. In Figure 2, regression a (β: −20.07, 95% CI (−26.18; −13.96) indicates that participating in aquatic exercise leads to a reduction in pain, and regression b (β: 0.46, 95% CI (0.11; 0.81) shows a direct relationship between the decrease in pain and depression levels as a result of participating in the program. The figure also shows the direct effect (β: −14.38, 95%CI (−24.08; −4.69] of the reduction of depression among the participants in the program. Lastly, the indirect effect (β: −9.36, 95%CI (−17.68, −2.66)), indicates that the improvements in pain levels can explain the decrease in depression. Thus, our mediation hypothesis was confirmed.

## 4. Discussion

To the best of our knowledge, no previous studies have examined the mediational role of improving bodily pain perception on depression after an exercise intervention in RA patients. The main findings in this study showed that the improvement in body pain mediated the effects of participating in the program on depression reduction. Our results suggest that physical exercise interventions in RA patients should emphasize progressively reducing pain perceptions and including physical exercises that do not cause pain. These findings could aid in exercise prescriptions and recommendations for RA patients and similar pathologies.

Although many studies show associations between pain and depression in people with RA and similar pathologies [12,27], this is the first longitudinal study examining whether pain is a mediator of depression. In line with this, there are studies that can strengthen our findings. It has been shown in population studies that the rates of depression increased during the postoperative period in patients who had not previously reported depression [28]. In a longitudinal study with a 12-year follow-up period, it was found that pain preceded the onset of depression; however, the researchers showed that depression did not predict the onset of pain [29]. Therefore, it seems that biological and social mechanisms behind this relationship can explain it. From the biological point of view, neuroimaging studies have shown changes in neuroplasticity (molecular, cellular, and synaptic processes that modify connectivity between neurons and neuronal circuits) in acute and chronic pain. Concretely, depressive changes in cerebral areas, such as the cortex and thalamus, have been identified in response to acute and chronic pain [30]. It has also been found that acute and chronic pain induces changes in molecular concentrations. Concretely, the studies showed increases in the proliferation of glutamate, neuropeptides, and neurotrophic factor as mediators of depression [17,30]. Further, analgesic drugs have been used to treat chronic, pain-induced depression. Concretely, opioids and benzodiazepines are behind the molecular changes associated with depression [17].

From the social perspective, experiencing pain leads to a cycle of impairments in activities of daily living and social relationships. It has been shown that the existence of pain generates catastrophism and kinesophobia [31], which leads to apathy toward performing activities of daily living and a more sedentary lifestyle [32], thus ushering in disability and poor quality of life [33].

Our results significantly improved pain perception and depression in the experimental group. Therefore, participating in an aquatic exercise program alleviates RA’s comorbidities. The mediational role explains that the decreased perception of pain partly causes the improvements in depression. As mentioned above, the improvement in depression could be because the participants in the water exercise program experienced a decrease in their perception of pain. The analgesia of exercise, per se, could cause this improvement in pain perception. In previous studies, it has been reported that chronic pain exercise evokes analgesia [13,14]. Further, hyperexcitability of the motor cortex has been documented in people with chronic pain [34]. Therefore, the improvements in pain in program participants could also be due to a modulation of pain induced by physical exercise, as has been indicated in previous studies [13,14,35].

A direct effect of participating in the aquatic physical exercises program was improving depression. Our results are comparable with previous studies based on physical exercise, which improved after training among depressed people [36,37]. Partially, one of the mechanisms that lowered the depression levels in the experimental group could be the socialization and fun experienced during the exercise sessions. In fact, a systematic review with meta-analysis of randomized controlled trials has demonstrated that physical exercise improves depression and anxiety in those with arthritis and other rheumatic conditions [11].

In our study, improvements in the physical fitness of the participants of the program were also demonstrated. We can consider this improvement as a logical effect of performing physical activity, and it is comparable to what has been found in similar studies of patients with chronic pain [38].

The present study presents limitations that require further discussion. First, the reduced size of the sample may have contributed to decreasing statistical power to detect changes. On the other hand, although this research provides information on the applicability and robustness of a water training program in female RA patients, the generalization of the results to populations of different ages, sexes, or diseases must be done with caution.

## 5. Conclusions

Women with RA who participated in the physical exercise aquatic program improved their physical fitness, pain, and depression levels. The reduction in pain mediated the effects of the program on depression. Consequently, physical exercise in water is recommended for women with RA.

## Figures and Tables

**Figure 1 ijerph-20-05872-f001:**
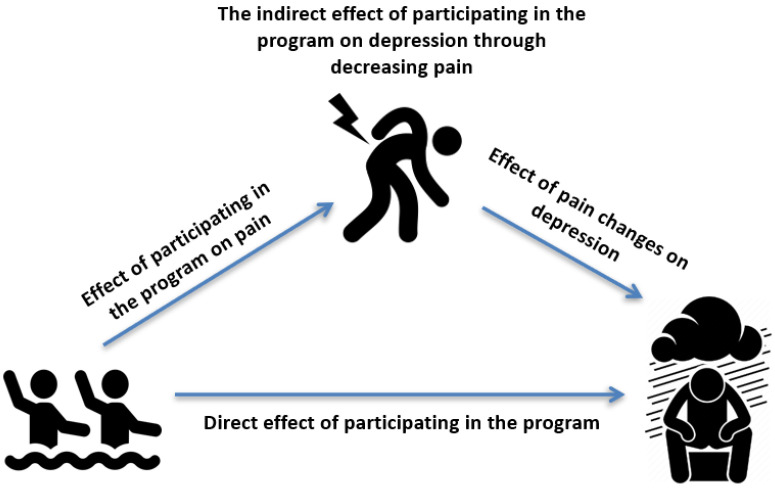
The panel of the mediation model.

**Figure 2 ijerph-20-05872-f002:**
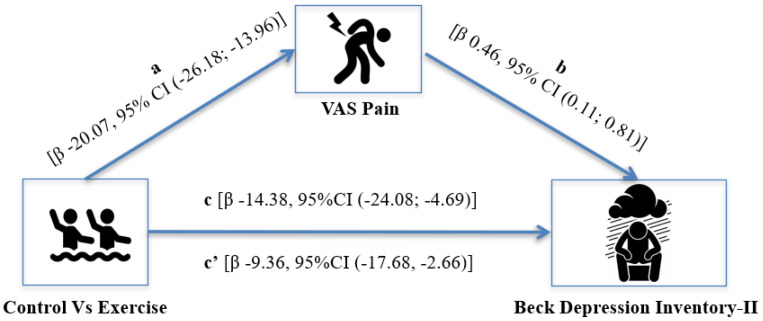
Mediation analysis of participating in an aquatic exercise program on depression (BDI-II) scores through pain (VAS), adjusted by age, self-reported physical activity, and body mass index. The number of bootstrap samples = 5000. The indirect effect is statistically significant at the 95% confidence interval (CI) when the CI does not include 0. Betas (β) are reported as the product of simultaneous regression with bootstrap replacement: path a = association between participating in the exercise program vs. non-participating and pain; path b = association between pain and depression; path c = direct effect; c’ = indirect effect.

**Table 1 ijerph-20-05872-t001:** Characteristics of study participants.

	Control Group	Exercise Group	*p*-Value
	n = 23	n = 21	
Age, years	56.2 ± 10.8	56.7 ± 11.2	0.892
Weight, kg	69.7 ± 9.5	69.1 ± 7.7	0.826
Height, m	1.5 ± 0.1	1.6 ± 0.1	0.330
BMI (kg/m^2^)	27.9 ± 3.5	27.1 ± 3.2	0.399

BMI, body mass index.

**Table 2 ijerph-20-05872-t002:** Effects on physical fitness, pain, and depression after 12 weeks of aquatic exercise.

	Baseline		Changes		Quantitative Chances as %			Clinical Inference
	CG (n = 23)	EG (n = 21)	Δ% (95%CI)	ES (95%CI)	Harmful	Trivial	Beneficial	
Physical Fitness								
Upper body strength (kg/kg)	0.23 ± 0.09	0.23 ± 0.08	10.9 (7.0 to 15.0)	0.27 (0.17 to 0.36)	0	2	98	very likely
Lower body power (W·kg^−1^)	2.11 ± 0.56	2.20 ± 0.75	4.0 (1.2 to 6.8)	0.12 (0.04 to 0.20)	3	97	0	very likely
Upper body mobility (cm)	−7.22 ± 7.30	−6.29 ± 8.92	8.9 (−10.7 to 32.8)	0.12 (−0.15 to 0.39)	1	72	27	possibly
Lower body mobility (cm)	−1.04 ± 2.87	−1.95 ± 4.46	6.5 (−8.8 to 24.6)	0.11 (−0.16 to 0.39)	1	73	26	possibly
Agility/dynamic balance (s)	6.47 ± 1.67	5.95 ± 1.67	4.8 (2.2 to 7.4)	0.18 (0.08 to 0.27)	0	70	30	possibly
Pain	4.83 ± 0.83	4.48 ± 0.60	−23.4 (−28.9 to −17.5)	1.72 (1.24 to 2.21)	0	0	100	most likely
Depression	16.91 ± 10.97	15.38 ± 10.26	−25.7 (−32.9 to −17.8)	0.45 (0.30 to 0.60)	0	0	100	most likely

Values expressed as mean ± SD, EG = experimental group, CG = control group, Δ% (95%CI).

## Data Availability

Not applicable.

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
