# Peer review of "Effects of Aquatic Exercises for Women with Rheumatoid Arthritis: A 12-Week Intervention in a Quasi-Experimental Study with Pain as a Mediator of Depression"

_ijerph, 2023, doi:10.3390/ijerph20105872_

Round 1

Reviewer 1 Report

Dear Authors,

I received a manuscript to review titled: Effects of aquatic exercises in women with rheumatoid arthritis: a 12-week intervention in a quasi-experimental study. Pain as the mediator of depression.

The study is written very meticulously, the methodology sufficiently explains the preparation of the study, the results are properly documented, and the discussion is sufficiently complemented by other studies. It is evident that the authors have a broad overview of the subject. I only have minor comments on the work.

I suggest providing the following modifications and additions:

1) In Chapter 2.1 „Design and participants“, you specify that one of the criteria was „no participation in a physical exercise program or no history of physical exercise in the last three months“(line 80).   In this regard, I am not clear about the sentence in lines 146-147 where you state: „after controlling for confounding variables such as age, self-reported physical activity and body mass index“.

Please clarify what you meant by "self-reported physical activity" if these participants were excluded from the study.

2) In the Chapter „Results“ you state in the very first sentence that you found statistically significant differences between groups in BMI.  But according to the results in Table 1, this does not seem to be the case.  Please clarify.

3) In the results in lines 163 and 167, you use the term "functional fitness," but this is not the correct term for the variables you are examining. I suggest you either use the term as stated in Chapter 2.2.2, or you may refine all the terms to "muscular fitness."

Author Response

Dear reviewer, we appreciate your comments.

1) We refer to the fact that the analyzes were executed adjusted by the physical activity carried out during the experimental phase of the study. With this, we ensure that the physical activity carried out by both groups during the experimental period does not alter the effects of the study. As we indicated, an exclusion criterion was to have carried out physical activity in the last three months before the start of the study.

self-reported physical activity has been changed to "physical activity

2) Corrected

3) We have rewritten all terms as "physical fitness"

Reviewer 2 Report

First of all, thank you for submitting to IJERPH.

Unfortunately, I have decided that it is unsuitable for publication for the following reasons.

A lot of research on aquatic exercise and depression has already been done. Are there any special reasons for RA patients? I did not find any characteristics in the author's study.

The age range of 18 to 80 years is too wide to study RA. In particular, RA increases with age and the incidence rate increases, and it is an extremely rare disease in the age of 10 to 20 years. In addition, in this study, the intervention was exercise, and fitness, depression, and pain were measured. All of these are greatly influenced by age.

The analysis is too monotonous to be published as research. Is there a reason why an independent t-test was not performed for comparison between groups in the two studies? Besides, if it's a normal distribution, is there a reason why you didn't do repeated two way ANOVA?

There are many limitations in the description of the document. In particular, research methods should be described in much more detail than now. Furthermore, the discussion should describe more objectively the differences between previous studies and this study.

Author Response

Dear reviewer, we appreciate your appreciation. Then we respond to your comments.

Water activities can be beneficial for people with rheumatoid arthritis. Rheumatoid arthritis is an autoimmune disease that causes inflammation in the joints, which can lead to pain, stiffness, and decreased range of motion. Water activities can help relieve pressure on joints and reduce pain, allowing people with rheumatoid arthritis to perform low-impact exercise.
Additionally, water provides gentle resistance, which can help strengthen your muscles without putting stress on your joints. Water activities can also help improve flexibility and mobility, which is particularly important for people with rheumatoid arthritis.
However, it is important to remember that each person is unique and may respond differently to water activities and, It can have a very beneficial effect on issues associated with depression and anxiety.

Arthritis can appear in children and adolescents between the ages of 10 and 18. This form of arthritis is known as juvenile idiopathic arthritis (JIA) and is one of the most common rheumatic conditions in children. JIA affects the joints, causing pain, stiffness, swelling, and decreased range of motion. There are different types of JIA, each with its own symptoms and characteristics. When this disease exists, anxiety and depression are always present, regardless of age, as the weaknesses are similar.

The analysis performed to know the treatment effects was ANCOVA adjusted for baseline. The ANCOVA approach answers a different research question: whether the post-test means, adjusted for pre-test scores, differ between the two groups. In the ANCOVA approach, the whole focus is on whether one group has a higher mean after the treatment. It’s appropriate when the research question is about the mean value at the end. Not about gains, growth, or changes. The adjustment for the pre-test score in ANCOVA has two benefits. One is to make sure that any post-test differences truly result from the treatment, and aren’t some left-over effect of (usually random) pre-test differences between the groups. The other is to account for variation around the post-test means that comes from the variation in where the patients started at pretest. So when the research question is about the difference in means at post-test, this is a great option. It’s very common in medical studies because the focus there is about the size of the effect of the treatment.

Round 2

Reviewer 2 Report

I have no further comments